# Deciphering the influence of *Bacillus subtilis* strain Ydj3 colonization on the vitamin C contents and rhizosphere microbiomes of sweet peppers

**Ying-Ru Liang[1,2], Fang-Chin Liao[2], Tzu-Pi Huang** ⬥[1,3]*

**1** Department of Plant Pathology, National Chung-Hsing University, Taichung, Taiwan, **2** Agricultural Chemicals and Toxic Substances Research Institute, Council of Agriculture, Executive Yuan, Taichung, Taiwan, **3** Innovation and Development Center of Sustainable Agriculture, National Chung Hsing University, Taichung, Taiwan

\* tphuang@nchu.edu.tw

**Data Availability Statement:** All relevant data are within the manuscript.

**Funding:** The research was financially supported by the Council of Agriculture, Taiwan [106AS-

## Abstract

*Bacillus subtilis* strain Ydj3 was applied to sweet peppers to understand the influence of this bacterium on the growth, fruit quality, and rhizosphere microbial composition of sweet pepper. The promotion of seed germination was observed for sweet pepper seeds treated with the Ydj3 strain, indicating that Ydj3 promoted seed germination and daily germination speed (131.5 ± 10.8 seeds/day) compared with the control (73.8 ± 2.5 seeds/day). Strain Ydj3 displayed chemotaxis toward root exudates from sweet pepper and could colonize the roots, which enhanced root hair growth. Following the one-per-month application of strain Ydj3 to sweet pepper grown in a commercial greenhouse, the yield, fruit weight, and vitamin C content significantly increased compared with those of the control. Additionally, the composition of the rhizosphere bacterial community of sweet pepper changed considerably, with the *Bacillus* genus becoming the most dominant bacterial genus in the treated group. These results suggested that *B. subtilis* Ydj3 promotes seed germination and enhances fruit quality, particularly the vitamin C content, of sweet pepper. These effects may be partly attributed to the *B. subtilis* Ydj3 colonization of sweet pepper roots due to Ydj3 chemotaxis toward root exudates, resulting in the modulation of the rhizosphere bacterial community.

## Introduction

Sweet pepper (*Capsicum annuum* L.) belongs to the *Capsicum* genus and the Solanaceae family and is among the most popular vegetables, has high nutrient contents, and is particularly rich in vitamins A and C. Antioxidant compounds, including water-soluble vitamin C and water-insoluble vitamin A and carotenoids, are abundant in pepper fruits and these vitamins play important roles due to their potent anti-inflammatory properties. Vitamin C has also been shown to act as a disease suppressant in humans. The production area of sweet pepper in Taiwan is approximately 2,498 hectares, which yields 29,221 tons of sweet pepper each year, and

12.4.1-PI-P1, 110AS-5.4.2-PI-P2], the Ministry of Science and Technology, Taiwan [MOST 110-2321-B-005-006; 109-2321-B-005-022; 109-2313-B-005-032], and the "Innovation and Development Center of Sustainable Agriculture, NCHU" from the Featured Area Research Center Program within the framework of the Higher Education Sprout Project by the Ministry of Education in Taiwan. The funders had no role in study design, data collection and analysis, decision to publish, or preparation of the manuscript.

**Competing interests:** The authors have declared that no competing interests exist.

the price of sweet pepper fluctuates between 43 and 85 New Taiwan dollars per kg. According to the FAO, more than 70% of the world's bell peppers are produced in Asia [1]. China is the largest producer of bell peppers, followed by Mexico and Indonesia [2]. The United States is the sixth largest and 363,647 hg/ha chilies and peppers were produced in 2020 [1, 2].

The qualities of pepper fruit are affected by the cultivar [3], agricultural practices [4, 5], fertilizer use [6], macronutrient and micronutrient application [7], disease control strategies, and postharvest conditions [8]. Changes in the soil bacterial community may enhance the suppression of Fusarium wilt disease in pepper [9]. Additionally, root colonization by *Bacillus subtilis* strain SL-44, which is attracted by the root exudates of pepper, has been reported to promote pepper plant growth [10].

The benefits of plant growth-promoting rhizobacteria (PGPR) on disease suppression and seed germination in pepper have been described previously [11, 12]. Cisternas-Jamet et al. [11] showed that the root inoculation of *Bacillus amyloliquefaciens* on green bell pepper affected the biochemical composition of fruits, including inducing changes in the calcium, iron, and vitamin C contents, especially when *B. amyloliquefaciens* was inoculated at the seedbed stage prior to transplantation. Furthermore, Mamphogoro et al. [13] revealed that fresh picked peppers are colonized by different bacterial communities on the fruit surface, and antagonistic bacterial communities that exist on the fruit surface can be found independent of the agronomic strategies employed during their growth. These bacterial communities may be important for peppers established in open fields, protecting them against abiotic and biotic stresses. However, no comprehensive assessment has examined the effects of PGPR treatment from seeds to fruits, and the diversity and dynamics of rhizosphere soil communities for fresh pepper plants grown under different treatment conditions have also not been examined. The objectives of this study were to evaluate the potential effects of the *Bacillus subtilis* strain Ydj3 on sweet pepper plants, from seeds to fruits, including effects on seed germination, seedling growth, fruit quality, and fruit yield, to reveal the interaction between the bacterial strain Ydj3 and sweet pepper root, and to understand the dynamics of the rhizosphere microbiome in the presence and absence of the Ydj3 strain.

## Material and methods

### Isolation, identification, and growth conditions of the bacterial strain

*Bacillus subtilis* strain Ydj3 was isolated from the rhizosphere of dragon Chinese juniper (*Juniperus chinensis* L. var. *kaizuka* Hort. ex Endl), in Yongjing Township, Changhua County, Taiwan (23˚91'93N, 120˚51'33 E), and was assessed for antagonistic activity against *Colletotrichum gloeospoioides*, *Botrytis cinerea*, and *Phytophthora capsici* by dual culture assay [14]. The species identity was characterized based on biochemical and physiological characteristics using the Biolog Gen III microbial identification system (Biolog Inc., CA, USA) and sequence analysis based on 16S rRNA [15] and *gyrB* sequences [16]. The 16S rRNA sequences of *B. subtilis* strain Ydj3 were deposited in the GenBank database under accession number MW911495; the *gyrB* sequences were deposited under accession number MW922029. *B. subtilis* strain Ydj3 was routinely maintained on Luria-Bertani (LB) agar plates (Difco, Detroit, MI, USA) and was cultured in LB broth (Difco) at 30˚C and 150 rpm for 24 h unless otherwise stated. The field research and materials collected were under the agreement by the farm owner and granted permission by the Council of Agriculture, Taiwan.

### Seed germination assay and seedling growth

Sweet pepper seeds (Yellow star, Known-You seed, No.SV-030, Kaohsiung, Taiwan) were surface-disinfected in 6% NaClO solution for 5 min and rinsed four times in sterile distilled

water. The disinfected seeds were completely dipped in a *B. subtilis* Ydj3 culture suspension at a final concentration of $1\times10^8$ CFU/mL and then transferred into a 128-cell plug tray containing 10% perlite peat soil (Jiffy, Moerdijk, Netherlands) and incubated in a growth chamber at 25˚C with a 12-h light/12-h dark cycle. The germination parameters and biomass were evaluated 6 weeks posttreatment. The germination percentage (GP), mean germination time (MGT), germination value (GV), and daily germination speed (DGV) were calculated based on the formulas described by Djavanshir and Pourbeik [17]. Each treatment was performed with three replicates, and each replicate contained 20 seeds.

For the seedling growth assay, sweet pepper seeds (yellow star, Known-You seed) were surface disinfected, as described above. Each germinated seed was transferred into a 3.5-inch-diameter pot containing 10% perlite peat soil (Jiffy) and incubated in a growth chamber at 25˚C with a 12-h light/12-h dark cycle. After emergence, each seedling was irrigated with 2 mL of *B. subtilis* Ydj3 culture broth, once per week for 5 weeks, at a final concentration of $1\times10^6$ CFU/mL. Distilled water and uninoculated LB broth were used as controls. Twenty seedlings were used for each treatment, with three replicates. After 40 days of treatment, the plants were harvested, and the dry weights of the aboveground and underground components of the plants were assessed.

## Field experiment and growth conditions

Sweet pepper (yellow star) seedlings were grown in a seedbed with 10% perlite peat soil (Jiffy) and then transplanted to a commercial greenhouse in Xinyi Township, Nantou County, Taiwan (23˚37'13.2"N, 120˚53'00.5"E). The experiment was conducted in a commercial greenhouse. Before transplanting the seedlings, peat soil was sanitized with hot water, and the treatment group was inoculated with *B. subtilis* Ydj3 culture broth at a final concentration of $1\times10^7$ CFU/mL. Tap water was used as a control. Pepper seedlings were transplanted into 10-m-long and 40-cm-wide bed tanks in double rows. Each row was approximately 30 cm apart, with 15 cm between each seedling in the row. Each treatment was performed with three replicates, and each replicate contained approximately 100 plants, which were drip-irrigated and fed liquid fertilizer (Nitrophoska®perfect, EuroChem, Mannheim, Germany) through the irrigation system. Fertilizer management was performed as recommended by The World Vegetable Center (AVRDC).

## Fruit quality measurement and vitamin C content

*B. subtilis* Ydj3 culture broth was applied by root irrigation once per month to sweet pepper plants grown in a commercial greenhouse, and tap water was used as a control as described above. The horticultural traits, including total harvest yield, fruit size, fruit weights, fruit firmness, total seed weight per fruit, total soluble solids (˚Brix), and vitamin C content, were recorded. The harvested fruit sizes were classified into large (>200 g/fruit), medium (100–200 g/fruit), and small (<100 g/fruit), based on the weight of each fruit. Total soluble solids were determined using a handheld refractometer (Brix 0–33%, Atago, Master-M, Tokyo, Japan). The vitamin C concentration in sweet pepper determined by measuring ascorbic acid using the ultraviolet (UV) spectroscopy method and 10 μg /mL ascorbic acid was used for standard curve preparation. Briefly, 20 mature pepper fruit samples were collected from plants in each treatment condition, 5 g from each sample was milled with 5 mL of 1% HCl solution, and the milled fruit juice was collected. A 2 mL sample of the milled juice was transferred into a 25-mL tube containing 8 mL of distilled water and shaken gently. Then, 0.2 mL of diluted sample was mixed with 0.4 mL of 10% HCl solution, distilled water was added to a total volume of 10 mL. The absorbance was measured at 243 nm using an ultraviolet–visible (UV–Vis)

spectrophotometer (BioRad SmartSpec™ 3000, Bio-Rad Laboratories, Hercules, California, U. S.A.). All determinations were performed in triplicate, and the results are expressed in μg vitamin C per g fresh fruit weight (μg Vit C/g). Vitamin C content was determined using the following equation:

$$\text{Vitamin C content } (\mu g/g) = (\mu \times v/v1 \times w) \tag{1}$$

where μ is the vitamin C concentration, as interpolated from the standard curve; v is the total sample volume; v1 is the volume used for spectrophotometer determination; and w is the blended weight of the sample (g).

## Microscopy observation and root colonization assay

The colonization assay was performed using a modified method described by Zhang et al. [18]. Sweet pepper seeds were surface-disinfected with 6% NaClO and seeded in 10% perlite peat soil (Jiffy) for 45 days. After removing the peat soil from the roots, the seedlings were transferred into 48 mL of sugar-free, half-strength Hoagland medium No. 2 (Sigma, Darmstadt, Germany). After 2 days of transplantation, 2 mL of *B. subtilis* Ydj3 culture broth at $1\times10^7$ CFU/mL was added to half-strength Hoagland medium No. 2. Distilled water and LB broth were used as controls. For each treatment, five plant roots were observed using optical microscopy and scanning electron microscopy. The plants were incubated in a growth chamber at 100 rpm and 25˚C for 24 h. Before microscopic observation, the roots were collected and gently washed twice with sterile water. The sweet pepper roots from plants grown under each treatment condition were observed using an optical microscope (Olympus Fluorescence BX60, Tokyo, Japan) and a scanning electron microscope (SEM, JEO JSM-6330F SEM, Tokyo, Japan). For optical microscope observation, at least ten root hairs from different zones for each treatment were randomly chosen and their lengths were measured. For SEM observations, the roots from plants grown in each treatment were dehydrated as follows: the root samples were gently washed twice with distilled water and completely dehydrated through incubation in an ascending ethanol series: 50%, 70%, 80%, 90%, and 95% for 30 min each and three times in 100% for 30 min each time. The colonization of bacterial cells on the roots was observed using a JEO JSM-6330F SEM at 2.80 kV, and images were obtained. Each sample was observed at 2000X, 3000X and 4000X magnification and at least 3 images were taken for analysis.

For the root colonization assay, a 0.1-g root sample from each treatment was ground with a pestle mill, 0.9 mL of sterile distilled water was added. The suspension was serially diluted and plated on an LB agar plate and cultured at 30˚C, 24 h to count the numbers of colonized cells. The *Bacillus* and other bacteria were differentiated by colony morphology on the LB agar plates, and *Bacillus*-like colonies were randomly picked for the identification of *Bacillus* species by 16S RNA sequencing as described above [15].

## Collection of root exudates from sweet pepper and chemotaxis assay

Sweet pepper seedlings were prepared as described above. For each treatment, 50 sweet pepper seedlings were used to collect root exudates for use in the chemotaxis assay. After removing the peat soil, the roots were gently washed in sterile distilled water four times. Seedlings with four leaves were transplanted into 50-mL tubes containing 50 mL of sterile, half-strength, sucrose-free Hoagland medium No. 2 (Sigma) at 28˚C for 5 days. After 5 days, the roots of the seedlings were washed with sterile distilled water to avoid nutrient contamination and transferred into 50-mL tubes containing 40 mL sterile distilled water in a growth chamber for 24 h (12-h light/12-h dark) at 100 rpm and 25˚C. The collected solution (approximately 2000 mL) containing root exudates was filtered through a 0.45-μm membrane filter (Millipore,

Darmstadt, Germany), concentrated by a vacuum concentrator (Büchi, vacuum controller B-721, Flawil, Switzerland) to a final volume of 50 mL, and stored at −80˚C for further investigation.

A modified capillary assay based on the procedure described by Mazumder et al. [19] was performed for the quantitative measurement of *B. subtilis* strain Ydj3 chemotaxis in response to root exudates collected from sweet pepper seedlings. Strain Ydj3 was grown in LB broth until it reached the logarithmic growth phase, as determined by an optical density at 600 nm ($OD_{600}$) of 0.8. The cells were collected by centrifugation, gently washed twice with chemotaxis buffer (20 μM EDTA, 100 mM potassium phosphate, pH 7.0), and resuspended in chemotaxis buffer to an $OD_{600}$ of 0.8. A disposable 2-cm 25-gauge needle loaded with 100 μL the concentrated root exudate was immersed in a 200-μL pipette tip containing 100 μL of cell suspension. After 30 min of incubation at room temperature, the contents of the needle were transferred into a sterile Eppendorf tube by syringe. The suspension was diluted $10^3$-, $10^4$-, and $10^5$-fold and plated on LB agar plates. Colony-forming units (CFUs) were counted after incubation for 24 h at 30˚C. The experiment was repeated three times, and each treatment included three replicates.

## Rhizosphere microbiome analysis

For the analysis of the soil microbial community composition for each treatment, untreated and *B. subtilis* Ydj3-treated rhizosphere soil samples were collected from sweet pepper roots at the beginning of planting and at the end of fruit harvest and stored at −20˚C until further analysis. Bulk peat soil samples were collected after hot water sanitization and before transplanting sweet pepper as controls (Ctrl 0). Rhizosphere soil samples without *B. subtilis* Ydj3 treatment were used as a second control (Ctrl 1). Approximately 5 g of peat soil from each soil sample was subjected to DNA extraction using QIAamp DNA Microbiome Kit (QIAGEN, Germantown, MD, USA), and another 5 g of peat soil was used for the analysis of the total bacterial population by the serial dilution and plating method to count cell numbers.

Microbial identification was performed using amplicon sequencing of 16S rRNA. PCR amplification was performed to amplify the V3–V4 conserved regions of bacterial 16S rRNA gene sequences. The V3 and V4 regions were amplified using a forward primer (5′ CCTAC GGRRBGCASCAGKVRVGAAT 3′) and a reverse primer (5′ GGACTACNVGGGTWTCTAA TCC 3′). Next-generation sequencing library preparation and Illumina MiSeq sequencing were conducted at GENEWIZ, Inc. (Suzhou, China). DNA samples were quantified using a Qubit 2.0 Fluorometer (Invitrogen, Carlsbad, CA, USA). Amplicons were generated from 30–50 ng DNA using a MetaVx™ Library Preparation kit (GENEWIZ, Inc., South Plainfield, NJ, USA).

DNA libraries were validated by an Agilent 2100 Bioanalyzer (Agilent Technologies, Palo Alto, CA, USA) and quantified by a Qubit 2.0 Fluorometer. DNA libraries were multiplexed and loaded onto an Illumina MiSeq instrument, according to the manufacturer's instructions (Illumina, San Diego, CA, USA). Sequencing was performed using a 2×300 paired-end configuration; image analysis and base calling were conducted using the MiSeq Control Software embedded in the MiSeq instrument.

The QIIME data analysis package was used for 16S rRNA data analysis. Quality filtering of joined sequences was performed, and sequences that did not fulfill the following criteria were discarded: sequence length <200 bp, no ambiguous bases, and mean quality score >20. The retained sequences were compared with the reference database (RDP Gold database) using the UCHIME algorithm to detect chimeric sequences, which were removed. The effective sequences were used in the final analysis. Sequences were grouped into operational taxonomic

units (OTUs) using the clustering program VSEARCH (1.9.6) against the Silva 119 database preclustered at 97% sequence identity. The Ribosomal Database Program (RDP) classifier was used to assign taxonomic categories to all OTUs at a confidence threshold of 0.8. The RDP classifier uses the Silva 132 database, which has taxonomic categories predicted to the species level.

## Statistical analysis

All data were analyzed by analysis of variance (ANOVA). Each treatment was performed using three replicates, and the results are expressed as the mean values. Significant differences among treatment means were analyzed using the least significant difference (LSD) test at $p < 0.05$ with IBM SPSS Statistics software, Version 22.0. (Armonk, NY, IBM Corp).

## Results

### Influence of *B. subtilis* Ydj3 application on seed germination and seedling growth

To evaluate the effects of *B. subtilis* Ydj3 application on seed germination and seedling growth, pepper seeds were dipped in *B. subtilis* Ydj3 broth culture at a final concentration of $10^8$ CFU/mL before transfer to peat soil. Seeds were treated with water and uninoculated LB broth as controls. Treatment with *B. subtilis* Ydj3 showed the highest GP (91.7 ± 7.2%) compared with the water (66.7 ± 6.8%) and LB broth (66.7 ± 9.0%) treatments (Table 1). The DGV following *B. subtilis* Ydj3 treatment (131.5 ± 10.8 seeds/day) also increased compared with water (73.8± 2.5 seeds/day) and LB broth (91.4 ± 8.6 seeds/day) treatments (Table 1). The application of *B. subtilis* Ydj3 broth culture also significantly promoted the daily germination speed (DGV) and shortened the MGT compared with those observed for the control treatments (Table 1). These results suggested that *B. subtilis* Ydj3 culture broth promoted the seed germination rate and percentage and reduced the germination time for sweet pepper.

For measures of seedling growth by irrigation with water, LB broth and *B. subtilis* Ydj3 culture broth, treatment with *B. subtilis* Ydj3 showed the highest shoot dry weight per plant (0.11 ± 0.04 g/plant), followed by treatment with LB broth (0.05 ± 0.04 g/plant) and water (0.04 ± 0.03 g/plant; Table 1). The shoot dry weight following *B. subtilis* Ydj3 treatment was enhanced by 275% compared with that of the water treatment control. However, no significant difference was found for the root dry weight between pepper seedlings treated with *B. subtilis* Ydj3 (0.02 ± 0.02 g/plant), LB broth (0.02 ± 0.02 g/plant), or water (0.03 ± 0.02 g/plant; Table 1). Our results indicated that the application of *B. subtilis* Ydj3 once per week to sweet pepper could stimulate shoot seedling growth.

**Table 1. Effects of *B. subtilis* Ydj3 colonization on seed germination and seedling growth of sweet peppers.**

| Treatment | Growth parameters | | | | | |
|---|---|---|---|---|---|---|
| | Germination percentage (GP) (%) | Mean germination time (MGT) (day) | Daily germination speed (DGV) (seeds/day) | Germination value (GV) | Root dry weight (g/plant) | Shoot dry weight (g/plant) |
| **dH₂O** | 66.7±6.8 a | 23.3±2.9 ab | 73.8±2.5 a | 2.9±0.4 a | 0.03±0.02 a | 0.04±0.03 c |
| **LB broth** | 66.7±9.0 a | 29.3±5.4 b | 91.4±8.6 b | 2.3±0.5 a | 0.02±0.02 a | 0.05±0.04 b |
| **Ydj3** | 91.7±7.2 b | 20.5±0.1 a | 131.5±10.8c | 4.5±0.5 b | 0.02±0.02 a | 0.11±0.04 a |

Means in the same column followed by the same letter are not significantly different at $P > 0.05$, according to one-way ANOVA and the least significant differences (LSD) test. Treatments: dH₂O, distilled water; LB broth, Luria-Bertani broth; Ydj3, *B. subtilis* Ydj3 culture broths at a final concentration of $1 \times 10^8$ CFU/mL.

**Table 2. Effects of *B. subtilis* Ydj3 colonization on the fruit qualities of sweet peppers.**

| Physicochemical composition of fruit | Treatment | |
|---|---|---|
| | Ctrl | Ydj3 |
| Total yield (kg) | 51607.7 ± 532.7 | 53903.3 ± 373.9 * |
| Mean number of large fruits | 74.0 ± 18.2 | 89.3 ± 5.9 |
| Weight of large fruits (g/fruit) | 241.1 ± 5.7 | 247.2 ± 5.1 * |
| Mean number of medium fruits | 198.7 ± 34.3 | 180.3 ± 8.5 |
| Weight of medium fruits (g/fruit) | 157.2 ± 3.2 | 164.9 ± 1.6 * |
| Mean number of small fruits | 36.7 ± 9.7 | 28.7 ± 6.2 |
| Weight of small fruits (g/fruit) | 74.1 ± 6.0 | 80.8 ± 12.0 |
| Mean weight of fruits (g/fruit) | 157.5 ± 5.0 | 164.3 ± 6.2 |
| Total seed weight (g/fruit) | 2.4 ± 0.1 | 2.5 ± 0.1 |
| Firmness [20] | 6.4 ± 0.1 | 6.4 ± 0.1 |
| TSS (˚Brix) | 5.6 ± 0.1 | 5.6 ± 0.1 |
| Vitamin C (μg/g) | 730.8 ± 14.4 | 790.9 ± 17.9 * |

Means in the same row followed by an asterisk (*) are significantly different at $P < 0.05$, according to one-way ANOVA and the least significant differences (LSD) test. TSS, total soluble solids.

## Influence of *B. subtilis* Ydj3 application on the physicochemical composition of sweet pepper fruits

The effects of *B. subtilis* Ydj3 application on fruit quality were assessed (Table 2). The results indicated that the total yield of plants treated with *B. subtilis* Ydj3 (53,903.3 ± 373.9 kg) was significantly higher than that of the control (51,607.7 ± 532.7 kg). The weight of large-sized fruit obtained from plants treated with *B. subtilis* Ydj3 (247.2 ± 5.1 g/fruit) was significantly ($p < 0.05$) heavier than that of large-sized fruit obtained from plants treated with water (241.1 ± 5.7 g/fruit); medium-sized fruit from *B. subtilis* Ydj3-treated plants (164.9 ± 1.6 g/fruit) was also significantly heavier than those from the water control group (157.2 ± 3.2 g/fruit). Although larger quantities of large fruits were collected from plants treated with *B. subtilis* Ydj3 (89.3 ± 5.9 mean number of large fruits) compared with those collected from control plants (74.0 ± 18.2 mean number of large fruits), the difference was not significant. The differences in numbers of medium- and small-sized fruits, weights of small-sized fruits, total average fruit weight, total seed weight, fruit firmness, and total soluble solids between *B. subtilis* Ydj3-treated plants and control plants were also not significant. However, sweet peppers collected from plants treated with *B. subtilis* Ydj3 contained significantly higher vitamin C contents (790.9 ± 17.9 μg/g) than those obtained from untreated control plants (730.8 ± 14.4 μg/g).

## Influence of *B. subtilis* Ydj3 application on root hair formation and root colonization

To investigate the effects of *B. subtilis* Ydj3 application on root hair growth and the root architecture of sweet peppers, the roots were examined by optical microscopy, and pictures were recorded, as shown in Fig 1. Roots treated with *B. subtilis* Ydj3 presented denser root hairs at apical meristem, elongation zone, and maturation zone than those in the control (Fig 1). Furthermore, the root hairs found in the apical meristem zone and the elongation zone (Fig 1D and 1E, respectively) were longer in the *B. subtilis* Ydj3-treated group than in the control group (Fig 1A and 1B, respectively). The lengths of root hairs in each treatment were measured using the software (cellSens Standard 1.9) installed in the Olympus Fluorescence BX60

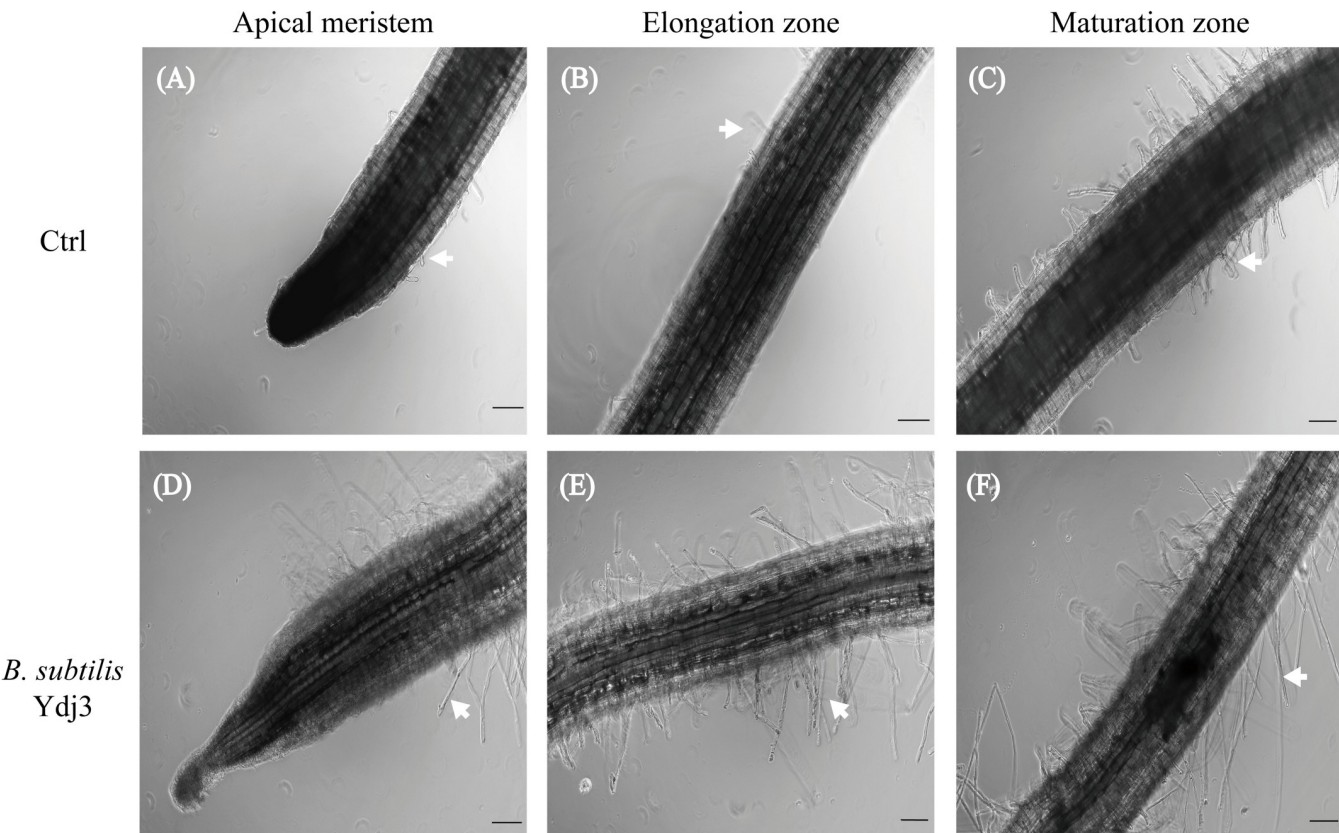

**Fig 1. Root architecture examined by optical microscopy.** (A, B, C) Control; (D, E, F) treatment with *B. subtilis* Ydj3. (A, D) Apical meristem; (B, E) elongation zone; (C, F) maturation zone of sweet pepper roots. Bar, 100 μm; arrow, root hair.

microscope, and results showed that the length of root hair of *B. subtilis* Ydj3 treatment had was longer than in the Ctrl treatment, with an average length in the apical meristem of 279.4 ± 177.3 μm compared with the Ctrl treatment, with an average length of 61.8 ± 8.8 μm; an average length of 375.9 ± 143.0 μm in the elongation zone, significantly longer than the Ctrl treatment, with an average length of 120.6 ± 63.4 μm; and an average length of 491.4 ± 21.9 μm in the maturation zone, significantly longer than the Ctrl treatment, with an average length of 172.4 ± 17.6 μm. These results suggested that *B. subtilis* Ydj3 applications enhanced root hair growth and changed the root architecture of sweet pepper.

To reveal the interaction between *B. subtilis* Ydj3 and the roots of sweet pepper, the colonization capability of strain Ydj3 was assessed by milling 0.1-g root samples, and plating the obtained solution on LB agar plates. The *Bacillus* and other bacteria were differentiated by colony morphology on the LB agar plate, and the identification of *Bacillus* species were confirmed by 16S RNA sequencing. The total bacterial population density in roots obtained from plants treated with strain Ydj3 was significantly higher ($6.84 \pm 0.04$ $\log_{10}$ CFU/g) than those detected on the roots of control plants treated with distilled water ($5.85 \pm 0.22$ $\log_{10}$ CFU/g) or LB broth ($6.66 \pm 0.10$ $\log_{10}$ CFU/g). While the bacteria obtained from plants treated with strain Ydj3 were all *Bacillus* species (data not shown).

Root colonization by *B. subtilis* Ydj3 was observed by SEM, and the control group revealed no bacterial cells on the root surface (Fig 2A). In contrast, *B. subtilis* Ydj3 cells were found to colonize the root surface and root hair (Fig 2B and 2C).

| Ctrl | *B. subtilis* Ydj3 | *B. subtilis* Ydj3 |

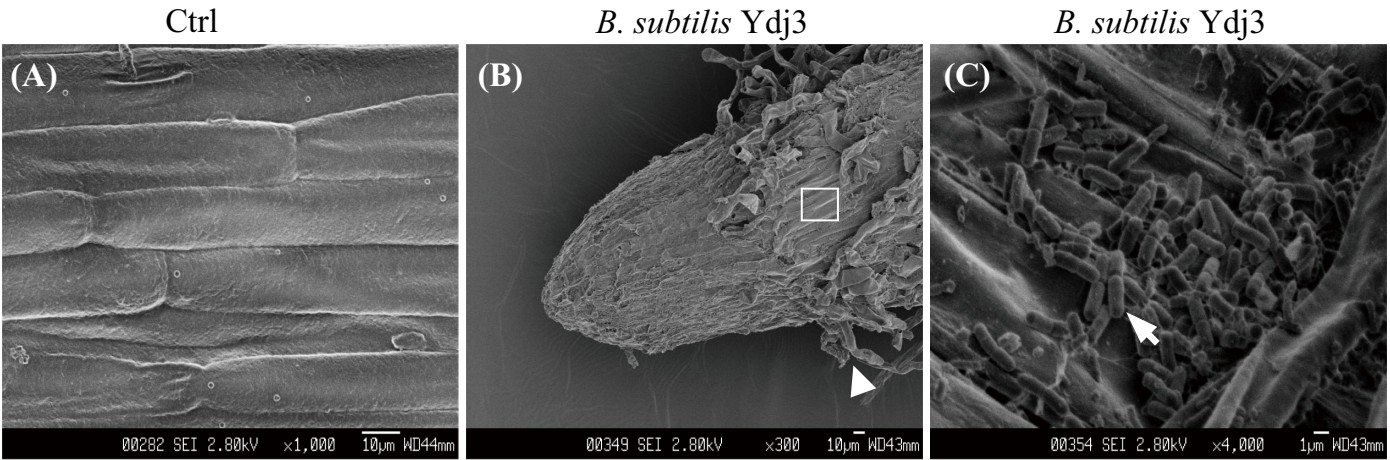

**Fig 2. Scanning electron microscopy photographs of sweet pepper roots.** (A) Treatment with distilled water as a control (Ctrl); (B, C) treatment with *B. subtilis* Ydj3. Bar, 10 μm. (A) plant cells were observed; (B) the apex of a root. Arrow head, root hairs; retangular area was enlarged in C; (C) Enlarged image of the retangukar area in B. Arrow head, bacteria.

### Chemotaxis to root exudate by *B. subtilis* Ydj3

The chemotactic effects of root exudates on *B. subtilis* Ydj3 were assessed using a modified capillary assay. The results showed that strain Ydj3 was chemoattracted by 40-fold concentrated root exudates, resulting in a population density of $4.53 \pm 0.08$ $\log_{10}$ CFU/mL compared with the distilled water control, which was characterized by a population density of $4.21 \pm 0.03$ $\log_{10}$ CFU/mL.

### Bacterial community composition in soil with and without *B. subtilis* Ydj3 application to sweet pepper

The analysis of the bacterial communities in the rhizosphere soil of sweet pepper plants treated with and without *B. subtilis* Ydj3 resulted in 171,368 bacterial reads, consisting of 546 OTUs (97% cutoff). The Q20 values (%) of Ctrl 0, Ctrl 1 and Ydj3 were 91.36%, 90.24% and 91.31%, respectively, and the Q30 values (%) of Ctrl 0, Ctrl 1 and Ydj3 were 87.10%, 85.4% and 86.90%, respectively. A total of 17 distinct bacterial phyla were detected among the samples, including 14 phyla, 130 genera, and 55 species in the peat soil before transplanting (Ctrl 0); 17 phyla, 153 genera, and 73 species in the rhizosphere soil (Ctrl 1); and 16 phyla, 158 genera, and 77 species in the sample with *B. subtilis* Ydj3 treatment (Ydj3). The most abundant phylum in all treatments was Actinobacteria, which constituted 35.6%, 15.3%, and 14.7% of the bacterial population detected in the Ctrl 0, Ctrl 1, and Ydj3 samples, respectively (Fig 3A). In the Ctrl 0 sample, the other abundant phyla were Proteobacteria (27.9%), Bacteroidetes (16.6%), and Firmicutes (7.1%). In the rhizosphere soil (Ctrl 1), the abundant phyla were Proteobacteria (42.7%), Actinobacteria (15.3%), and Firmicutes (12.0%); in the sample treated with *B. subtilis* Ydj3 (Ydj3), the abundant phyla were Proteobacteria (35.0%), Firmicutes (22.5%), and Actinobacteria (14.7%; Fig 3A). Except for unclassified genera, *Bacillus* was the most abundant (16.9%) of the top 17 genera identified in the Ydj3 treatment group and represented 7.0% and 16.9% of the total bacterial populations of Ctrl 0 and Ctrl 1, respectively (Fig 3B). The enriched bacterial genera identified in the pepper rhizosphere between Ctrl 1 and Ydj3 included *Nocardioides* (1.3% to 1.7%), *Bacillus* (7.0% to 16.9%), and *Gemmatimonas* (5.82% to 5.87%).

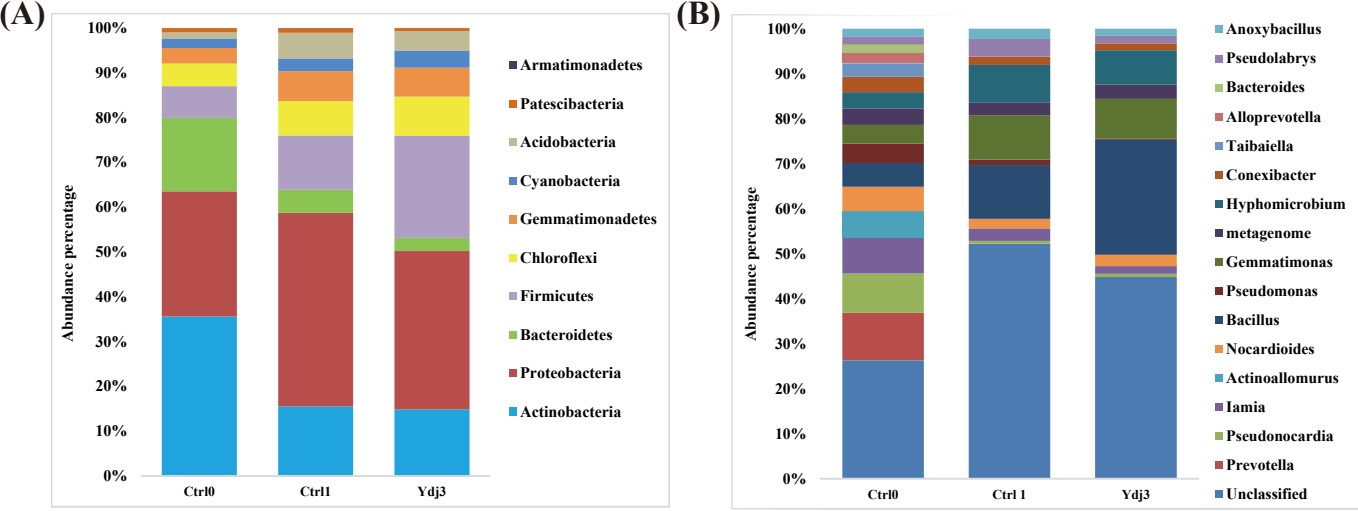

**Fig 3. Composition of the bacterial community in rhizosphere soils with and without *B. subtilis* Ydj3 treatment.** (A) Relative abundances of most abundant phyla; Ctrl 0, bulk peat soil treated with hot water before sweet pepper transplant; Ctrl 1, rhizosphere soil without *B. subtilis* Ydj3 treatment, 6 months after sweet pepper transplant; Ydj3, rhizosphere soil with *B. subtilis* Ydj3 treatment, 6 months after sweet pepper transplant. (B) Relative abundances of the top 17 genera.

## Discussion

Microbial seed coating is an efficient technology for the safe and effective delivery of beneficial microbes to plant roots or the rhizosphere. Li and Hu [21] found that exogenous treatments with *B. subtilis* QM3 applied to wheat seeds increased β-amylase activities and promoted the germination and seedling survival rates [21]. Another study by Tu et al. [22] indicated that the use of *B. subtilis* SL-13 at $10^6$ CFU/seeds as a microbial seed coating agent could promote seed germination and biomass in cotton [22]. Our results showed that seeds dipped in a *B. subtilis* Ydj3 cell suspension at $10^8$ CFU/mL promoted the germination of sweet pepper seeds compared with the untreated control. Additionally, the application of *B. subtilis* Ydj3 broth culture at $10^8$ CFU/mL resulted in a 25% increase in seed germination and a 2.5-fold increase in shoot dry weight compared with the average value of control treatments. Our results suggested that *B. subtilis* Ydj3 displayed chemotaxis in response to root exudates produced by sweet pepper, was capable of colonizing sweet pepper roots, and induced root hair growth, which may be beneficial for the improved nutrient absorption of sweet pepper plants, resulting in the promotion of plant growth. Whether β-amylase production by *B. subtilis* Ydj3 is involved in the promotion of seed germination remains to be determined. Additionally, according to Vacheron et al. [23] and Grover et al. [24], understanding and quantifying the impact of PGPR on roots and the whole plant remain challenging. Several studies have indicated that PGPR modifies the root structure, including primary root growth, lateral root density, and root dry weight [24]. However, the root dry weight was not enhanced in our results. This might be because the plants were young and the variation remained nonsignificant.

The rhizosphere plays an important role in plant–microbe interactions. López-Bucio et al. [25] revealed that *Bacillus megaterium* inoculation inhibited primary root growth followed by increases in the lateral root number, lateral root growth, and root hair length in *Arabidopsis thaliana* [25]. Their results suggested that plant growth promotion and root architectural alterations induced by *B. megaterium* may involve auxin- and ethylene-independent mechanisms [25]. Additionally, Vacheron et al. [23] indicated that many PGPR affect root system architecture via modulation of host gormonal balance, including the plant auxin pathway. *Bacillus*

*amyloliquefaciens* T-5 exhibited a chemotactic response to the malic acids found in root exudates produced by tomato plants, which induced biofilm formation and the colonization of tomato roots by *B. amyloliquefaciens* T-5 [26]. Similarly, our results indicated that *B. subtilis* Ydj3 could be chemotactically attracted by root exudates produced by sweet pepper, colonize the root surface, and alter the root architecture by increasing root hair growth. Our unpublished data showed that *B. subtilis* Ydj3 produces the plant growth regulator indole-3-acetic acid (17 mg/L), which may also contribute to alterations in root architecture and induce plant growth.

Various rhizobacteria are known to promote plant growth, improve plant yields, and alter fruit quality. The inoculation of tomato roots with *B. subtilis* BEB-l3bs was demonstrated to enhance the marketable yield, fruit weight and length, and the texture of red fruits [27]. Rahman et al. [28] reported improvements in plant growth, yield, various antioxidant contents (carotenoids, flavonoids, phenolics, and total anthocyanins), and total antioxidant activities in strawberry fruits following the application of both *B. amyloliquefaciens* BChi1 and *Paraburkholderia fungorum* BRRh-4 compared with nontreated controls [28]. The application of a strain from the genus *Phyllobacterium* has been reported to significantly increase the vitamin C contents of strawberry [29]. The application of microbial fertilizer products increased the flavonoid and lignin contents in *Arabidopsis* leaves through the induction of phenylpropanoid pathway genes [30]. In our study, the inoculation of sweet pepper plants with *B. subtilis* Ydj3 significantly increased the shoot dry weight of seedlings, the total yield, the weights of large- and medium-sized sweet pepper fruits, and the vitamin C contents of the fruits. Whether the observed increase in the vitamin C contents of sweet pepper induced by *B. subtilis* Ydj3 treatment is associated with any changes in the phenylpropanoid pathway remains to be investigated.

Plant species have been hypothesized to accommodate specific compositions of rhizosphere bacterial communities to optimize growth and prevent infection by pathogens. The use of biofertilizer may shift the composition of the plant microbiome. Sun et al. [31] reported that the application of *B. subtilis* biofertilizer reduced the abundance of *ureC* genes and increased the abundance of functional genes (bacterial *amoA* and comammox *amoA*) and ammonia-oxidizing bacteria, suggesting that the conversion of fertilizer nitrogen to $NH_4^+$-N decreased, and the nitrification process increased. Thus, *B. subtilis* biofertilizer is considered an effective control strategy for agricultural $NH_3$ emissions and maintaining high crop yields. Additionally, their results indicated that Actinobacteria was the dominant phylum in the soil before fertilization, whereas Proteobacteria became the new dominant phylum during the stable period [31]. Potential PGPR bacteria (*Bacillus*), nitrobacteria (*Nitrospira*), and photosynthetic bacteria (*Rhodoplanes*) were found to be enriched in *B. subtilis* biofertilizer compared with fertilizer without *B. subtilis* [31]. Qiao et al. [32] reported a similar finding for a single time point, whereas Sun et al. [31] examined a quantification curve after applying *B. subtilis* biofertilizer throughout the pak choi growth period. Their findings also indicated that the application of *B. subtilis* PTS-394 did not result in a permanent effect on the rhizosphere microbial community, suggesting the compatibility of *B. subtilis* PTS-394 with the environment [32]. Similar to the data presented by Sun et al. [31], the dominant phylum identified in bulk peat soil (Ctrl 0) in our study was Actinobacteria. After a 6-month growth period for sweet pepper, the most abundant phyla shifted to Proteobacteria in the untreated control group and to Firmicutes in the *B. subtilis* Ydj3-treated group. Additionally, the *Bacillus* species abundance was enriched in the *B. subtilis* Ydj3 treatment group compared with the control group.

In conclusion, the application of *B. subtilis* Ydj3 promoted seed germination, seedling vigor, and fruit production in sweet pepper and enhanced the antioxidant vitamin C contents of sweet pepper fruit. The efficacy of bacterial treatment for plant growth promotion and the

enhancement of fruit quality may be partly attributed to the chemotactic response of *B. subtilis* Ydj3 to sweet pepper root exudates, and the colonization of sweet pepper roots alters the root architecture. Additionally, the application of *B. subtilis* Ydj3 shifted the bacterial community compositions of the sweet pepper rhizosphere. Our findings suggested that *B. subtilis* Ydj3 could be used as a bioagent for the sustainable production of high-quality sweet pepper.

## Acknowledgments

We thank Drs. Jenn-Wen Huang and Yi-Hsien Lin for their valuable comments and helpful discussions.

## Author Contributions

**Conceptualization:** Ying-Ru Liang, Tzu-Pi Huang.

**Data curation:** Ying-Ru Liang, Tzu-Pi Huang.

**Formal analysis:** Ying-Ru Liang, Fang-Chin Liao, Tzu-Pi Huang.

**Funding acquisition:** Ying-Ru Liang, Tzu-Pi Huang.

**Investigation:** Ying-Ru Liang, Fang-Chin Liao, Tzu-Pi Huang.

**Methodology:** Ying-Ru Liang, Tzu-Pi Huang.

**Project administration:** Ying-Ru Liang, Tzu-Pi Huang.

**Resources:** Ying-Ru Liang, Tzu-Pi Huang.

**Software:** Fang-Chin Liao, Tzu-Pi Huang.

**Supervision:** Tzu-Pi Huang.

**Validation:** Ying-Ru Liang, Tzu-Pi Huang.

**Visualization:** Ying-Ru Liang, Tzu-Pi Huang.

**Writing – original draft:** Ying-Ru Liang.

**Writing – review & editing:** Tzu-Pi Huang.

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
