## [Decision Letter · Decision Letter 0]

19 Nov 2021

PONE-D-21-26845Deciphering the influence of Bacillus subtilis strain Ydj3 colonization on the vitamin C contents and rhizosphere microbiomes of sweet peppersPLOS ONE

Dear Dr. Huang,

Thank you for submitting your manuscript to PLOS ONE. After careful consideration, we feel that it has merit but does not fully meet PLOS ONE’s publication criteria as it currently stands. Therefore, we invite you to submit a revised version of the manuscript that addresses the points raised during the review process.

We look forward to receiving your revised manuscript.

Kind regards,

Ying Ma, Ph.D.

Academic Editor

PLOS ONE

Journal Requirements:

Whilst you may use any professional scientific editing service of your choice, PLOS has partnered with both American Journal Experts (AJE) and Editage to provide discounted services to PLOS authors. Both organizations have experience helping authors meet PLOS guidelines and can provide language editing, translation, manuscript formatting, and figure formatting to ensure your manuscript meets our submission guidelines. To take advantage of our partnership with AJE, visit the AJE website (http://aje.com/go/plos) for a 15% discount off AJE services. To take advantage of our partnership with Editage, visit the Editage website (www.editage.com) and enter referral code PLOSEDIT for a 15% discount off Editage services.  If the PLOS editorial team finds any language issues in text that either AJE or Editage has edited, the service provider will re-edit the text for free.

A clean copy of the edited manuscript (uploaded as the new *manuscript* file)”"

3. In your Methods section, please provide additional location information, including geographic coordinates of your field collection (for dragon Chinese juniper rhizosphere) if available.

4. In your Methods section, please provide additional details regarding the dragon Chinese juniper used in your study, including ensuring you have described the source. For more information regarding PLOS' policy on materials sharing and reporting, see https://journals.plos.org/plosone/s/materials-and-software-sharing#loc-sharing-materials.

The research was financially supported by the Council of Agriculture, Taiwan [106AS-12.4.1-PI-P1, 110AS-5.4.2-PI-P2], the Ministry of Science and Technology, Taiwan [MOST 110-2321-B-005-006; 109-2321-B-005-022; 109-2313-B-005-032], and the “Innovation and Development Center of Sustainable Agriculture, NCHU” from the Featured Area Research Center Program within the framework of the Higher Education Sprout Project by the Ministry of Education in Taiwan.

Reviewers' comments:

Reviewer's Responses to Questions

**Comments to the Author**

1. Is the manuscript technically sound, and do the data support the conclusions?

Reviewer #1: Partly

Reviewer #2: Yes

Reviewer #3: Yes

2. Has the statistical analysis been performed appropriately and rigorously? 

Reviewer #1: N/A

Reviewer #2: Yes

Reviewer #3: Yes

3. Have the authors made all data underlying the findings in their manuscript fully available?

Reviewer #1: Yes

Reviewer #2: Yes

Reviewer #3: Yes

4. Is the manuscript presented in an intelligible fashion and written in standard English?

Reviewer #1: No

Reviewer #2: Yes

Reviewer #3: Yes

5. Review Comments to the Author

Reviewer #1: It is a relatively useful article on the study of probiotics "Bacillus subtilis strain Ydj3", and has a positive effect on the sweet pepper industry. The paper needs to be reconsidered after modification. Please see the comments showed as follows:

Line 89: add citation ” assessed for antagonistic activity against Colletotrichum gloeospoioides, Botrytis cinerea, and Phytophthora capsici by dual culture assay.”

Lines 220-223: Supplementary information on where the soil sample was collected, was it collected from the surface?

Fig. 1: Redraw the figure to facilitate understanding of groups and parts. Besides, the length of root hairs needs to be quantified.

Lines 342-348: Combined with the last part of the results, there may be a large number of other bacteria in the root system, so LB is not suitable for counting bacillus.

Lines 349-351: Due to the small sampling area of SCANNING electron microscopy, whether this result is repeated for many times. After all, a large number of bacteria were also detected in the last results (line 347: 5.85 ± 0.22 log10 CFU/g)

Fig. 2: Redraw to help understand groups. In addition, does the bacteria on the root surface caused pathological change?

Lines 364-385:

1. Please add more quality control information about 16s-hiseq.

2. The form of FIG. 3 can only show the average gap between groups, but cannot show the inter-sample gap (standard deviation) within groups. Please add relevant information.

3. Whether similar results of ” antagonistic activity against Colletotrichum gloeospoioides, Botrytis cinerea, and Phytophthora capsici by dual culture assay (in the section materials and methods)” can be obtained through 16s-hiseq.

Reviewer #2: The manuscript describes a comprehensive study on the effects Bacillus subtilis strain Ydj3 on seed germination, growth, fruit quality and rhizosphere microbial composition of sweet pepper. The authors obtained evidences for the promotion of seed germination, shoot biomass, fruit yield, weight and vitamin C content upon inoculation. These effects were partly attributed to chemotaxis of Ydj3 towards root exudates, resulting in the modulation of the bacterial community in rhizosphere. The work is well structured and is presented clearly. Techniques and methodologies are suitable and well described. The results are well presented and analyzed. It is my opinion that this is an important study for this area of research and for regular readers of this journal. I have just a few minor comments and suggestions for corrections:

Line 25 – I don’t think the term “priming” is adequate here. Please rephrase.

Line 108 – Please include the abbreviation of germination value (GV).

Line 118 and 120 – Seedlings are young plants, and so after 40 days of treatment the correct designation should be “plants”.

Line 139 – As described above, correct?

Line 165 – Zhang et al. (2014) is reference [27].

Lines 225-226 – Is this a repetition of the above? (lines 221-222)?

Line 278 – Did you mean germination value (instead of daily germination speed)?

Line 283-284 (Table 1) – Please indicate units for mean germination time, germination value and daily germination speed. Correct the abbreviation of daily germination speed (DGV, not DGP). Are columns “Germination value” and “Daily germination speed” swapped?

Line 304-305 – These are the total fruit yield values, not by group, right? If yes, remove /group.

Line 320 (Table 2) – Total yield: Same as in the previous comment. Large fruits, medium fruits, small fruits: did you mean number of fruits/group (average of three replicates)? Lee et al. is reference [12].

Line 331 – showed that the distribution

Line 338 (Fig 1) – Are the captions of (A,D) and (C,F) swapped?

Line 368 – “A total of 16 distinct bacterial phyla”: Is this correct number of distinct phyla?

Line 404 – 25% increase in seed germination: how was this value obtained and what parameter does it refer to?

Line 405 -… compared with the average value of control treatments

Line 388 (Fig 3 caption) – (A) Relative abundances of most abundant phyla?

Reviewer #3: The manuscript titled “Deciphering the influence of Bacillus subtilis strain Ydj3 colonization on the vitamin C contents and rhizosphere microbiomes of sweet peppers” studies the influence of the Bacillus subtilis strain Ydj3 on the growth, fruit quality, and rhizosphere microbial composition of sweet peppers. Work presents an issue of economic importance due to the use of a bacteria as bioagent which produces high-quality of sweet pepper and deserves to be investigated. The manuscript is short, precise and well written. Authors presents results of original research and the rest of the requirements of the journal are fulfilled.

Below, I list some suggestions for authors:

Introduction

Page 5, line 55: Perhaps the authors can add more up-to-date citations

Material and Methods

-Microscopy observation and root colonization, Page 11 and 12

Please add number of replicates, number of plants per replicate, number of analyzed images.

Results

Page 21, line 338: In the description of figure 1, the letters are wrong, (A, D) is the meristematic zone and (C, F) is the zone of differentiation.

Commonly, the hairy area is called the differentiation zone, perhaps a more in-depth study of the state of differentiation of cells in these areas that present root hairs with treatment with the bacteria should be carried out. Perhaps the meristematic and elongation zones are more apical.

Page 22, line 353: A more detailed description of figure 2 would be desirable, indicating that it is observed in the images.

In image A, cells are observed, in B, the apex of a root (it would be desirable to indicate the bacteria) and in C, the root hairs (it would be desirable to indicate in the image which are the root hairs and which it's bacteria).

Discussion

If possible, explain in more detail the non-variation of the dry weight of the root between treatments (table 1), although the architecture of the root varies, increasing the root hairs in those roots that are treated with the bacteria.

6. PLOS authors have the option to publish the peer review history of their article (what does this mean?). If published, this will include your full peer review and any attached files.

Reviewer #1: No

Reviewer #2: No

Reviewer #3: No

---

## [Author Response · Author response to Decision Letter 0]

3 Jan 2022

PONE-D-21-26845. 

"Deciphering the influence of Bacillus subtilis strain Ydj3 colonization on the vitamin C contents and rhizosphere microbiomes of sweet peppers"

04-January-2022

Dear Dr. Ma and reviewers,

Thank you and the reviewer for the valuable comments to improve the manuscript! 

We have addressed you and the reviewer’s comments in black text below, with our responses in blue. Line numbers noted below correspond to the revised manuscript file. We also include a revised manuscript text with track changes in the resubmission. Additionally, the manuscript was edited for proper English language, spelling and grammar by native English speaking editors at American Journal Experts (AJE), and the certificate was also included.

Response to Editor:

“**3. In your Methods section, please provide additional location information, including geographic coordinates of your field collection (for dragon Chinese juniper rhizosphere) if available.**”

We greatly appreciate you and the reviewer to point out what we have missed from the previous review, we had added the geographic information (23°91'93N, 120°51'33 E) of field collection in the description (line 83).

“**4. In your Methods section, please provide additional details regarding the dragon Chinese juniper used in your study, including ensuring you have described the source.**”

We didn’t use the dragon Chinese juniper in our study, we collected the soil from the rhizosphere of the dragon Chinese juniper. 

“**5. Thank you for stating the following financial disclosure: 

The research was financially supported by the Council of Agriculture, Taiwan [106AS-12.4.1-PI-P1, 110AS-5.4.2-PI-P2], the Ministry of Science and Technology, Taiwan [MOST 110-2321-B-005-006; 109-2321-B-005-022; 109-2313-B-005-032], and the “Innovation and Development Center of Sustainable Agriculture, NCHU” from the Featured Area Research Center Program within the framework of the Higher Education Sprout Project by the Ministry of Education in Taiwan. Please state what role the funders took in the study. If the funders had no role, please state: "The funders had no role in study design, data collection and analysis, decision to publish, or preparation of the manuscript." If this statement is not correct you must amend it as needed. Please include this amended Role of Funder statement in your cover letter; we will change the online submission form on your behalf.

Thank you for the instruction of the statement about the financial support. We have put the statement "The funders had no role in study design, data collection and analysis, decision to publish, or preparation of the manuscript." as suggested in the cover letter.

“6. Please include your full ethics statement in the ‘Methods’ section of your manuscript file. In your statement, please include the full name of the IRB or ethics committee who approved or waived your study, as well as whether or not you obtained informed written or verbal consent. If consent was waived for your study, please include this information in your statement as well. “

The statement “The field research and materials collected were under the agreement by the farm owner and granted permission by the Council of Agriculture, Taiwan.” was included in the “Method” section. (Lines 92-93)

Response to Reviewer #1:

“Line 89: add citation ” assessed for antagonistic activity against Colletotrichum gloeospoioides, Botrytis cinerea, and Phytophthora capsici by dual culture assay.”

The reference was cited in the main text (line 85) and included in the reference list [2]. 

“Lines 220-223: Supplementary information on where the soil sample was collected, was it collected from the surface?”

We collected the soil samples from the roots of sweet pepper. Some of the fibrous root will be cut and removed, and then the soil samples were collected. 

The sentence was revised as “For the analysis of the soil microbial community composition for each treatment, untreated and B. subtilis Ydj3-treated rhizosphere soil samples were collected from the sweet pepper roots at the beginning of planting and at the end of fruit harvest and stored at −20 °C until further analysis.” (Lines 213-216)

“Fig. 1: Redraw the figure to facilitate understanding of groups and parts. Besides, the length of root hairs needs to be quantified.”

The figure was labeled as suggested, and the length of the root hairs was quantified and the descriptions were stated in the Material and Methods and Result sections as follows.

In the Material and Methods section, the sentences “For optical microscope observation, at least ten root hairs from different zones for each treatment were randomly chosen and their lengths were measured.” were added (lines 168 to 170).

In the Results section, the sentences “The lengths of root hairs in each treatment was measured using the software (cellSens Standard 1.9) installed in the Olympus Fluorescence BX60 microscope, and results showed that the length of root hair of B. subtilis Ydj3 treatment had was longer than in the Ctrl treatment, with an average length in the apical meristem of 279.4 ± 177.3 µm compared with the Ctrl treatment , with an average length of 61.8 ± 8.8 µm; an average length of 375.9 ± 143.0 µm in the elongation zone, significantly longer than the Ctrl treatment, with an average length of 120.6 ± 63.4 µm; and an average length of 491.4 ± 21.9 µm in the maturation zone, significantly longer than the Ctrl treatment, with an average length of 172.4 ± 17.6 µm.” were added. (lines 318 - 326)

“Lines 342-348: Combined with the last part of the results, there may be a large number of other bacteria in the root system, so LB is not suitable for counting bacillus.”

We agree with the reviewer that there are a large number of bacteria in the root system. The Bacillus and other bacteria were differentiated by colony morphology on the LB agar plates as shown in the following figure and the Bacillus-like colonies were randomly picked for the identification by 16S RNA sequencing. We found that the colonies obtained from the treatment with B. subtilis Ydj3 were all Bacillus species. The material and method and the statement in the result section was revised in the text (lines 180-182 and 336-342, respectively). 

(A) (B)

Fig. The colonies obtained from root samples of (A) the control; and (B) the B. subtilis Ydj3 treatment.

“Lines 349-351: Due to the small sampling area of SCANNING electron microscopy, whether this result is repeated for many times. After all, a large number of bacteria were also detected in the last results (line 347: 5.85 ± 0.22 log10 CFU/g)”

As stated in the above reply, the Bacillus and other bacteria were differentiated by colony morphology on the LB agar plates, and the Bacillus-like colonies were randomly picked for the identification of Bacillus species by 16S RNA sequencing.

“Fig. 2: Redraw to help understand groups. In addition, does the bacteria on the root surface caused pathological change?”

We have labeled respective treatment in the figure as suggested. We did not observe pathological effect on the root caused by B. subtilis Ydj3, a plant growth rhizobacterium (PGPR). The change in root architecture- enhancing root hair formation may be partly due to the production of indole-3-acetic acid (17 mg/L) by B. subtilis Ydj3 (our unpublished data and discussed in lines 423-425). Similarly, Richardson et al. (2009) indicated that PGPR could change the root architecture via production of plant hormone.

“Lines 364-385:

1. Please add more quality control information about 16s-hiseq.”

The cutoff value of the bacterial communities in the rhizosphere soil of sweet pepper plants is 97%, and the Effective Tags Q20 (%) and Q30 (%) values of each treatment showed below were all above 85%. 

treatment Q20 (%) Q30 (%)

Ctrl 0 91.36 87.10

Ctrl 1 90.24 85.40

Ydj3 91.31 86.90

Additionally, the description of the quality control was also stated in the main text as follows: “The Q20 values (%) of Ctrl 0, Ctrl 1 and Ydj3 were 91.36%, 90.24% and 91.31%, respectively, and the Q30 values (%) of Ctrl 0, Ctrl 1 and Ydj3 were 87.10%, 85.4% and 86.90%, respectively.” (Lines 362 - 365).

“2. The form of FIG. 3 can only show the average gap between groups, but cannot show the inter-sample gap (standard deviation) within groups. Please add relevant information.”

The main focus of the study was to compare the compositions of bacterial community between the control and the B. subtilis Ydj3 treatment, thus the inter-sample gap (standard deviation) within groups were not provided.

“3. Whether similar results of ” antagonistic activity against Colletotrichum gloeospoioides, Botrytis cinerea, and Phytophthora capsici by dual culture assay (in the section materials and methods)” can be obtained through 16s-hiseq.”

Thanks for the suggestion. The effect on the compositions of fungal community by the B. subtilis Ydj3 treatment will be investigated in the future.

Response to Reviewer #2:

“Line 25 – I don’t think the term “priming” is adequate here. Please rephrase.”

The term “ priming” was deleted as suggested. The statement was revised as “The promotion of seed germination was observed for sweet pepper seeds treated with the Ydj3 strain, indicating that Ydj3 promoted seed germination and daily germination speed (131.5 ± 10.8 seeds/day) compared with the control (73.8 ± 2.5 seeds/day).” (Lines 22 - 24)

“Line 108 – Please include the abbreviation of germination value (GV).”

The abbreviation of germination value (GV) was added. (Line 103)

“Line 118 and 120 – Seedlings are young plants, and so after 40 days of treatment the correct designation should be “plants”.

Line 139 – As described above, correct? ”

They were revised as suggested. (Lines 112 and 113)

“Line 165 – Zhang et al. (2014) is reference [27].”

The reference was cited in the main text (Line 156) and the reference list no [32] was revised accordingly. 

“Lines 225-226 – Is this a repetition of the above? (lines 221 - 222)?”

The repeated sentence was deleted. 

“Line 278 – Did you mean germination value (instead of daily germination speed)?”

Our results indicated that the B. subtilis Ydj3 treatment exhibited higher daily germination speed (DGV) than the control, so that the mean germination value (MGT) was shortened. (Lines 260-266)

“Line 283-284 (Table 1) – Please indicate units for mean germination time, germination value and daily germination speed. Correct the abbreviation of daily germination speed (DGV, not DGP). Are columns “Germination value” and “Daily germination speed” swapped?”

It was revised as suggested and the unit of each term was added in Table 1. (Table 1)

The unit of mean germination time (MGT) is day, and there is no unit of germination value (GV), GV is an index and calculated by the formula cited in reference no [6]. The unit of daily germination speed (DGV) is seeds/day, and the abbreviation was also revised. 

“Line 304-305 – These are the total fruit yield values, not by group, right? If yes, remove /group.”

It was revised as suggested and the term “/group” was deleted. (Lines 297 - 298)

“Line 320 (Table 2) – Total yield: Same as in the previous comment. Large fruits, medium fruits, small fruits: did you mean number of fruits/group (average of three replicates)? Lee et al. is reference [12].”

It was revised as suggested. In Table 2, the unit of total yield was kg, the unit of large fruits, medium fruit and small fruits was revised as “numbers of fruits”. The reference was added in Table 2 and the reference list no [16] (Line 305).

“Line 331 – showed that the distribution”

The description was deleted. (Line 313)

“Line 338 (Fig 1) – Are the captions of (A,D) and (C,F) swapped?”

The figure and figure legend were revised as suggested. (A, D) are apical meristem, (C, F) are maturation zone. (Fig 1.) (Lines 330-332)

“Line 368 – “A total of 16 distinct bacterial phyla”: Is this correct number of distinct phyla?”

It was revised as “ A total of 17 distinct bacterial phyla.” (Line 365)

“Line 404 – 25% increase in seed germination: how was this value obtained and what parameter does it refer to?”

The values was obtained as follows: GP (%) of Yjd3 – GP (%) of Ctrl =91.7% - 66.7% = 25%. (Line 398)

“Line 405 -… compared with the average value of control treatments”

The sentence was revised as suggested. (Lines 399-400)

“Line 388 (Fig 3 caption) – (A) Relative abundances of most abundant phyla?”

The Fig 3 caption was revised as suggested. (Line 382)

Response to Reviewer #3:

“Page 5, line 55: Perhaps the authors can add more up-to-date citations”

The reference was updated to the statistical data in 2020 in the main text (Line 52) and the reference list no [8] (Lines 498-499).

“Material and Methods

Microscopy observation and root colonization, Page 11 and 12

Please add number of replicates, number of plants per replicate, number of analyzed images.”

For the assay, each treatment contained 5 plants and at least 3 images were taken at different magnification for each sample. 

The statement was revised as the follows accordingly.

“For each treatment, five plant roots were observed using optical microscopy and scanning electron microscopy. “ (Lines 162-163) 

“For optical microscope observation, at least ten root hairs from different zones for each treatment were randomly chosen and their lengths were measured.” (Lines 168 – 170) 

“Each sample was observed at 2000X, 3000X and 4000X magnification and at least 3 images were taken for analysis.” (Lines 175 - 176)

“Results

Page 21, line 338: In the description of figure 1, the letters are wrong, (A, D) is the meristematic zone and (C, F) is the zone of differentiation.

Commonly, the hairy area is called the differentiation zone, perhaps a more in-depth study of the state of differentiation of cells in these areas that present root hairs with treatment with the bacteria should be carried out. Perhaps the meristematic and elongation zones are more apical.”

The legend of Fig 1 was corrected as suggested. (A,D), apical meristem; (B, E) elongation zone; (C,F) maturation zone. (Lines 330-332)

“Page 22, line 353: A more detailed description of figure 2 would be desirable, indicating that it is observed in the images.

In image A, cells are observed, in B, the apex of a root (it would be desirable to indicate the bacteria) and in C, the root hairs (it would be desirable to indicate in the image which are the root hairs and which it's bacteria).”

It was revised as suggested (Lines 347-350). The revised image was attached with the revised file. 

“Discussion

If possible, explain in more detail the non-variation of the dry weight of the root between treatments (table 1), although the architecture of the root varies, increasing the root hairs in those roots that are treated with the bacteria.”

The discussion was added as stated “understanding and quantifying the impact of PGPR on roots and the whole plant remain challenging. Several studies indicated that PGPR modified the root structure including primary root growth, lateral root density, root dry weight, however, the root dry weight was not enhanced in our results. This might be because the plants were young and the variation remained non-significant.” (Lines 406 - 410)

Thank you again for giving us the opportunity to address the question! 

Sincerely,

Tzu-Pi 

Tzu-Pi Huang, Ph. D.

Professor, Department of Plant Pathology 

Director, Pesticide Residue Analysis Center

National Chung Hsing University

145, Xing-Da Road, Taichung 40227, Taiwan

TEL: 886-4-22840780 ext. 379; 886-4-22859750

FAX: 886-4-22859750

E-mail: tphuang@nchu.edu.tw

---

## [Decision Letter · Decision Letter 1]

1 Feb 2022

PONE-D-21-26845R1Deciphering the influence of Bacillus subtilis strain Ydj3 colonization on the vitamin C contents and rhizosphere microbiomes of sweet peppersPLOS ONE

Dear Dr. Huang,

Thank you for submitting your manuscript to PLOS ONE. After careful consideration, we feel that it has merit but does not fully meet PLOS ONE’s publication criteria as it currently stands. Therefore, we invite you to submit a revised version of the manuscript that addresses the points raised during the review process.

We look forward to receiving your revised manuscript.

Kind regards,

Ying Ma, Ph.D.

Academic Editor

PLOS ONE

Journal Requirements:

Reviewers' comments:

Reviewer's Responses to Questions

**Comments to the Author**

1. If the authors have adequately addressed your comments raised in a previous round of review and you feel that this manuscript is now acceptable for publication, you may indicate that here to bypass the “Comments to the Author” section, enter your conflict of interest statement in the “Confidential to Editor” section, and submit your "Accept" recommendation.

Reviewer #2: (No Response)

Reviewer #3: All comments have been addressed

2. Is the manuscript technically sound, and do the data support the conclusions?

Reviewer #2: Yes

Reviewer #3: Yes

3. Has the statistical analysis been performed appropriately and rigorously? 

Reviewer #2: Yes

Reviewer #3: Yes

4. Have the authors made all data underlying the findings in their manuscript fully available?

Reviewer #2: Yes

Reviewer #3: Yes

5. Is the manuscript presented in an intelligible fashion and written in standard English?

Reviewer #2: Yes

Reviewer #3: Yes

6. Review Comments to the Author

Reviewer #2: The authors answered the questions and made most of the corrections suggested in the previous review. However, some minor aspects still need correction:

Section "Fruit quality and vitamin C content" (Lines 130-132): Field experiment and growth conditions were described in the previous section. Therefore, "as described above" should be added at the end of the first sentence of this section, or it seems to refer to another experiment.

Table 1. The position of the columns DGV and GV has been changed, but the contents are as in the previous version and are not in agreement with the text (lines 263-264). Please correct.

Table 2. The text in the first column should be unified. I suggest using "Number of large fruits" or "Mean number of large fruits" instead of "Large fruits (mean numbers of fruits)", and "Weight of large fruits" instead of "Large fruit weight". The same applies to medium fruits and small fruits. This will make Table 2 easier to read.

Lines 314-316 – Suggestion: “Roots treated with B. subtilis Ydj3 presented denser root hairs at apical meristem, elongation zone, and maturation zone than those in the control (Fig 1).”

Reviewer #3: The authors responded to the comments exhaustively justifying them and taking into account the suggestions of the reviewers.

In any case, I do not agree with the concept that there are root hairs in all root zones (apical, elongation and differentiation/maturation), according to what is incorporated in the new version of the work (lines

318-326).

Root hairs are found in the zone of differentiation/maturation only. Perhaps the authors will find bibliography that supports their findings.

7. PLOS authors have the option to publish the peer review history of their article (what does this mean?). If published, this will include your full peer review and any attached files.

Reviewer #2: No

Reviewer #3: **Yes: **María Victoria Rodriguez

---

## [Author Response · Author response to Decision Letter 1]

7 Feb 2022

PONE-D-21-26845. 

"Deciphering the influence of Bacillus subtilis strain Ydj3 colonization on the vitamin C contents and rhizosphere microbiomes of sweet peppers"

07-Febuary-2022

Dear Dr. Ma and reviewers,

Thank you and the reviewer for the valuable comments to improve the manuscript! 

We have addressed the reviewer’s comments in black text below, with our responses in blue. Line numbers noted below correspond to the revised manuscript file. Additionally, a revised manuscript text with track changes and an unmarked version were included in the resubmission. 

“Journal Requirements:

Please review your reference list to ensure that it is complete and correct. If you have cited papers that have been retracted, please include the rationale for doing so in the manuscript text, or remove these references and replace them with relevant current references. Any changes to the reference list should be mentioned in the rebuttal letter that accompanies your revised manuscript. If you need to cite a retracted article, indicate the article’s retracted status in the References list and also include a citation and full reference for the retraction notice.”

We also have reviewed and checked again the reference list and confirmed all the references cited are correct and have not been retracted. The reference cited in the statement “…fertilizer use [11], macronutrient and…” was changed to “…fertilizer use [10], macronutrient and…” (line 54).

Response to Reviewer #2:

“**1. Section "Fruit quality and vitamin C content" (Lines 130-132): Field experiment and growth conditions were described in the previous section. Therefore, "as described above" should be added at the end of the first sentence of this section, or it seems to refer to another experiment.**”

We greatly appreciate you and the reviewer to point out what we have missed from the previous review, we had added the description “as described above” at the end of the sentence (line 132-133).

“**2. Table 1. The position of the columns DGV and GV has been changed, but the contents are as in the previous version and are not in agreement with the text (lines 263-264). Please correct.**”

We had revised the data in the text respectively to Table 1 (line 263-264).

“**3. Table 2. The text in the first column should be unified. I suggest using "Number of large fruits" or "Mean number of large fruits" instead of "Large fruits (mean numbers of fruits)", and "Weight of large fruits" instead of "Large fruit weight". The same applies to medium fruits and small fruits. This will make Table 2 easier to read. **”

It was revised as suggested. (Lines 298-299 and line 307 Table 2)

“**4. Lines 314-316 – Suggestion: “Roots treated with B. subtilis Ydj3 presented denser root hairs at apical meristem, elongation zone, and maturation zone than those in the control (Fig 1). **”

It was revised as suggested (lines 316-317).

Response to Reviewer #3:

“**In any case, I do not agree with the concept that there are root hairs in all root zones (apical, elongation and differentiation/maturation), according to what is incorporated in the new version of the work (lines 318-326).

Root hairs are found in the zone of differentiation/maturation only. Perhaps the authors will find bibliography that supports their findings. **”

Thank you again for giving us the opportunity to address the question. Root hairs are cylindrical extensions of root epidermal cells that are important for acquisition of nutrients, microbe interactions, and plant anchorage. As descripted by Grierson et al. in the Arabidopsis Book (2014 (12); doi: 10.1199/tab.0172), root hair development is regulated by auxin and ethylene, especially by auxin homeostasis. Increased auxin or ethylene signaling resulted in moving the initiation site of root hair growth to a more apical position and increasing the amount of elongation during root hair tip growth. Decreased auxin or ethylene signaling showed the opposite effects. Additionally, as stated in the discussion section of our manuscript, Vacheron et al. indicated that many PGPR affect root system architecture via modulation of host gormonal balance, including the plant auxin pathway (lines 418-420). In our study, we observed the formation of root hairs initiated at apical zone, however, the mechanisms underlying the interaction between the B. subtilis Ydj3 treatment and the root hairs growth of sweet peppers remain to be investigated in the future. 

Thank you again for giving us the opportunity to address the questions! 

Sincerely,

Tzu-Pi 

Tzu-Pi Huang, Ph. D.

Professor, Department of Plant Pathology 

Director, Pesticide Residue Analysis Center

National Chung Hsing University

145, Xing-Da Road, Taichung 40227, Taiwan

TEL: 886-4-22840780 ext. 379; 886-4-22859750

FAX: 886-4-22859750

E-mail: tphuang@nchu.edu.tw

---

## [Editor Report · Decision Letter 2]

8 Feb 2022

Deciphering the influence of Bacillus subtilis strain Ydj3 colonization on the vitamin C contents and rhizosphere microbiomes of sweet peppers

PONE-D-21-26845R2

Dear Dr. Huang,

We’re pleased to inform you that your manuscript has been judged scientifically suitable for publication and will be formally accepted for publication once it meets all outstanding technical requirements.

Kind regards,

Ying Ma, Ph.D.

Academic Editor

PLOS ONE
---

## [Editor Report · Acceptance letter]

10 Feb 2022

PONE-D-21-26845R2 

Deciphering the influence of *Bacillus subtilis* strain Ydj3 colonization on the vitamin C contents and rhizosphere microbiomes of sweet peppers 

Dear Dr. Huang:

I'm pleased to inform you that your manuscript has been deemed suitable for publication in PLOS ONE. Congratulations! Your manuscript is now with our production department. 

Kind regards, 

on behalf of

Dr. Ying Ma 

Academic Editor

PLOS ONE